# Walking on the Edge: Fast, Low-Distortion Adversarial Examples

## Abstract

Adversarial examples of deep neural networks are receiving ever increasing attention because they help in understanding and reducing the sensitivity to their input. This is natural given the increasing applications of deep neural networks in our everyday lives. When white-box attacks are almost always successful, it is typically only the distortion of the perturbations that matters in their evaluation.

In this work, we argue that speed is important as well, especially when considering that fast attacks are required by adversarial training. Given more time, iterative methods can always find better solutions. We investigate this *speed-distortion trade-off* in some depth and introduce a new attack called *boundary projection* (BP) that improves upon existing methods by a large margin. Our key idea is that the classification boundary is a manifold in the image space: we therefore quickly reach the boundary and then optimize distortion on this manifold.

## 1 Introduction

*Adversarial examples* (Szegedy et al., 2013) are small, usually imperceptible perturbations of images or other data (Carlini & Wagner, 2018) that can arbitrarily modify a classifier's prediction. They have been extended to other tasks like object detection or semantic segmentation (Xie et al., 2017), and image retrieval (Li et al., 2018; Tolias et al., 2019). They are typically generated in a *white-box* setting, where the attacker has full access to the classifier model and uses gradient signals through the model to optimize for the perturbation. They are becoming increasingly important because they reveal the *sensitivity* of neural networks to their input (Simon-Gabriel et al., 2018; Fawzi et al., 2016; Amsaleg et al., 2017) including trivial cases (Azulay & Weiss, 2018; Engstrom et al., 2017) and they easily *transfer* between different models (Moosavi-Dezfooli et al., 2016a; Tramèr et al., 2017b).

Adversarial examples are typically evaluated by *probability of success* and *distortion*. In many cases, white-box attacks have probability of success near one, then only distortion matters, as a (weak) measure of imperceptibility and also of the ease with which adversarial samples can be detected. The *speed* of an attack is less frequently discussed. The fast single-step FGSM attack (Goodfellow et al., 2014) produces high-distortion examples where adversarial patterns can easily be recognized. At the other extreme, the *Carlini & Wagner* (C&W) attack (Carlini & Wagner, 2017), considered state of the art, is notoriously expensive. *Decoupling direction and norm* (DDN) (Rony et al., 2018) has recently shown impressive progress in distortion but mostly in speed.

Speed becomes more important when considering *adversarial training* (Goodfellow et al., 2014). This defense, where adversarial examples are used for training, was in fact introduced in the same work as FGSM. The latter remains the most common choice for generating those examples because of its speed. However, unless a powerful iterative attack is used (Madry et al., 2017), adversarial training is easily broken (Tramèr et al., 2017a).

In this work, we investigate in more depth the *speed-distortion trade-off* in the regime of probability of success near one. We observe that iterative attacks often oscillate across the classification boundary, taking long time to stabilize. We introduce a new attack that rather walks *along* the boundary. As a result, we improve the state of the art in distortion while keeping iterations at a minimum.

**Illustrating the attacks.** To better understand how our attack works, we illustrate it qualitatively against a number of existing attacks in Fig. 1. On this toy 2d classification problem, the class boundary and the path followed by the optimizer starting at input x can be easily visualized.

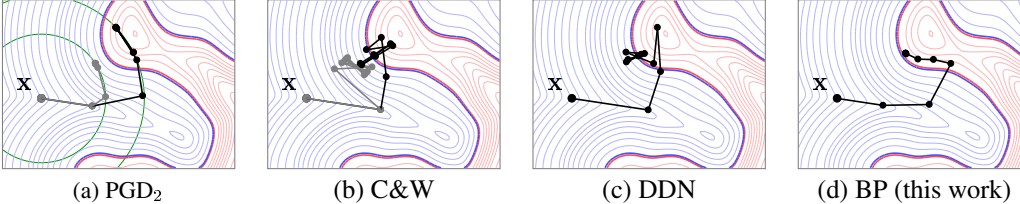

|  |  |  |  |
|---|---|---|---|
| (a) PGD$_2$ | (b) C&W | (c) DDN | (d) BP (this work) |

Figure 1: Adversarial attacks on a binary classifier in two dimensions. The two class regions are shown in red and blue. Contours indicate class probabilities. The objective is to find a point in the red (*adversarial*) region that is at the minimal distance to input **x**. Gray (black) paths correspond to low (high) distortion target $\epsilon$ for PGD$_2$ (Kurakin et al., 2016) (a, in green) or parameter $\lambda$ for C&W (Carlini & Wagner, 2017) (b). The simulation is only meant to illustrate basic properties of the methods. In particular, it does not include Adam optimizer (Kingma & Ba, 2015) for C&W.

PGD$_2$, an $\ell_2$ version of I-FGSM (Kurakin et al., 2016), a.k.a. PGD (Madry et al., 2017), is controlled by a distortion target $\epsilon$ and eventually follows a path on a ball of radius $\epsilon$ centered at **x** (*cf*. Fig. 1(a)). Section 4 shows that varying $\epsilon$ is an effective yet expensive strategy. It can only be done for a limited set of values so that the optimal distortion target per image may only be found by luck.

C&W (Carlini & Wagner, 2017) depends on a parameter $\lambda$ that controls the balance between distortion and classification loss. A low value may lead to failure. A higher value may indeed reach the optimal perturbation, but with oscillations across the class boundary (*cf*. Fig. 1(b)). Therefore, an expensive line search over $\lambda$ is performed internally.

DDN (Rony et al., 2018) (*cf*. Fig. 1(c)) increases or decreases distortion on the fly depending on success and at the same time pointing towards the gradient direction. It arrives quickly near the optimal perturbation but still suffers from oscillations across the boundary.

On the contrary, *boundary projection* (BP), introduced in this work (*cf*. Fig. 1(d)), cares more about quickly reaching the boundary, not necessarily near the optimal solution, and then walks *along the boundary*, staying mostly in the *adversarial* (red) region. It therefore makes steady progress towards the solution rather than going back and forth.

**Our attack.** Our key idea is that, once we reach the adversarial region near the boundary, the problem becomes *optimization on a manifold* (Absil et al., 2009): in particular, minimization of the $\ell_2$ distortion on a *level set* of the classification loss. When in the adversarial region, we project the distortion gradient on the *tangent space* of this manifold. We do this simply by targeting a particular reduction of the distortion while moving orthogonally to the gradient of the classification loss.

**Our benchmark.** Quantization is another major issue in this literature. Most papers implicitly assume that the output of a white-box attack is a matrix where pixel values are real numbers in $[0, 1]$. Rony et al. (2018) is one of the rare works where the output is a quantized. We agree with this definition of the problem. Indeed, an adversarial image is above all an image. The goal of an attacker is to publish images deluding the classifier (for instance on the web), and publishing implies compliance with pixels encoded in bytes.

**Contributions.** We make the following contributions. To our knowledge, we are the first to

1. Study optimization on the *manifold* of the classification boundary for an adversarial attack, providing an *analysis* under a number of constraints, such as staying on the tangent space of the manifold and reaching a distortion target.
2. Investigate theoretically and experimentally the quantization impact on the perturbation.
3. Achieve at the same speed as I-FGSM (Kurakin et al., 2016) (20 iterations) and under the constraint of a quantization, less distortion than state-of-the-art attacks including DDN, which needs 100 iterations on ImageNet.

## 2 PROBLEM, BACKGROUND AND RELATED WORK

### 2.1 PROBLEM FORMULATION

**Preliminaries.** Let $\mathcal{X} := \{0, \Delta, \ldots, 1 - \Delta, 1\}^n$ with $\Delta := 1/(L-1)$ denote the set of grayscale *images* of $n$ pixels quantized to $L$ levels, and let $\hat{\mathcal{X}} := [0,1]^n$ denote the corresponding real-valued images. An image of more than one color channels is treated independently per channel; in this case $n$ stands for the product of pixels and channels. A *classifier* $f : \hat{\mathcal{X}} \to \mathbb{R}^k$ maps an image $\mathbf{x}$ to a vector $f(\mathbf{x}) \in \mathbb{R}_+^c$ representing probabilities per class over $c$ given classes. The parameters of the classifier are not shown here because they remain fixed in this work. The classifier *prediction* $\pi : \hat{\mathcal{X}} \to [c] := \{1, \ldots, c\}$ maps $\mathbf{x}$ to the class label having the maximum probability:

$$\pi(\mathbf{x}) := \arg\max_{k \in [c]} f(\mathbf{x})_k. \tag{1}$$

If a *true label* $t \in [c]$ is known, the prediction is *correct* if $\pi(\mathbf{x}) = t$.

**Problem.** Let $\mathbf{x} \in \mathcal{X}$ be a given image with known true label $t$. An *adversarial example* $\mathbf{y} \in \mathcal{X}$ is an image such that the *distortion* $\|\mathbf{x} - \mathbf{y}\|$ is small and the probability $f(\mathbf{y})_t$ is also small. This problem takes two forms:

1. *Target distortion, minimal probability*:

$$\min_{\mathbf{y} \in \mathcal{X}} f(\mathbf{y})_t \tag{2}$$

$$\text{subject to} \quad \|\mathbf{x} - \mathbf{y}\| \leq \epsilon, \tag{3}$$

   where $\epsilon$ is a given distortion *target*. The performance is then measured by the *probability of success* $P_{\text{suc}} := \mathrm{P}(\pi(\mathbf{y}) \neq t)$ as a function of $\epsilon$.

2. *Target success, minimal distortion*:

$$\min_{\mathbf{y} \in \mathcal{X}} \|\mathbf{x} - \mathbf{y}\| \tag{4}$$

$$\text{subject to} \quad \pi(\mathbf{y}) \neq t. \tag{5}$$

   The performance is then measured by the *expected distortion* $\overline{D} := \mathrm{E}(\|\mathbf{x} - \mathbf{y}\|)$.

This work focuses on the second form, but we present example attacks of both forms in section 2.2.

**Untargeted attack.** The constraint $\pi(\mathbf{y}) \neq t$ in (5) is referred to as an *untargeted* attack, meaning that $\mathbf{y}$ is misclassified regardless of the actual prediction. As an alternative, a *targeted* attack requires that the prediction $\pi(\mathbf{y}) = t'$ is a target label $t' \neq t$. We focus on the former.

**Loss function.** We focus on a *white-box* attack in this work. Such an attack is specific to $f$, which is public. In this setting, attacks typically rely on exploiting the gradient of some loss function, using variants of gradient descent. A *classification loss* is defined on the probability vector $\mathbf{p} = f(\mathbf{y})$ with respect to the true label $t$. For an untargeted attack, this is typically the negative of cross-entropy

$$\ell(\mathbf{p}, t) := \log p_t. \tag{6}$$

We should warn that, while the cross-entropy is appropriate for bringing examples into the region of class $t$ during classifier training, its negative (6) is in general *not* appropriate for pulling them out during an attack. This is because this function is mostly flat in the class region. A common solution is to *normalize* the gradient of $\ell$ (Goodfellow et al., 2014; Rony et al., 2018), assuming it is nonzero. We consider more options in this work. A targeted attack on the other hand may use $-\log p_{t'}$, which works fine because it *brings* examples into class $t'$ region.

**Distortion.** This work focuses on the 2-norm $\|\cdot\|$ as a measure of distortion. Alternatives like 1-norm and $\infty$-norm are also common (Goodfellow et al., 2014; Carlini & Wagner, 2017). It is known that none is appropriate for measuring the imperceptibility of adversarial attacks, while more sophisticated measures like structural similarity (SSIM) (Wang et al., 2004) are limited too (Sharif et al., 2018). Measuring imperceptibility is arguably as difficult as classification itself.

**Integral constraint.** The constraint $\mathbf{y} \in \mathcal{X}$ in (3) and (5) is typically relaxed to $\mathbf{y} \in \hat{\mathcal{X}}$ during optimization. Some works conclude the attack by loosely quantizing the optimal solution onto $\mathcal{X}$,

typically by *truncation* towards zero. To our knowledge, DDN (Rony et al., 2018) is the only work to do *rounding* instead, and at the end of each iteration. Quantization is becoming an important issue in adversarial examples because the distortions achieved in recent papers are so small that quantization impacts a lot the perturbations. Appendix A provides a more in-depth study of the impact of the quantization.

## 2.2 ATTACKS

**Target Distortion**. Given a distortion target $\epsilon$, the *fast gradient sign method* (FGSM) (Goodfellow et al., 2014) performs a single step in the opposite direction of the (element-wise) sign of the loss gradient with $\infty$-norm $\epsilon$,

$$\mathbf{y} := \mathbf{x} - \epsilon \operatorname{sign} \nabla_{\mathbf{x}} \ell(f(\mathbf{x}), t). \tag{7}$$

This is the fastest method for problem (2)-(3). In the same work adversarial training was introduced, this method quickly generates adversarial examples for training. However, the perturbations are usually high-distortion and visible. The *iterative*-FGSM (I-FGSM) (Kurakin et al., 2016) initializes $\mathbf{y}_0 := \mathbf{x}$ and then iterates

$$\mathbf{y}_{i+1} := \operatorname{proj}_{B_\infty[\mathbf{x};\epsilon]}(\mathbf{y}_i - \alpha \operatorname{sign} \nabla_{\mathbf{x}} \ell(f(\mathbf{y}_i), t)), \tag{8}$$

where projection[1] is element-wise to the closed $\infty$-norm ball $B_\infty[\mathbf{x}; \epsilon]$ of radius $\epsilon$ and center $\mathbf{x}$, and also to $\hat{\mathcal{X}}$ (element-wise clipping to interval $[0, 1]$). This method is also known as *basic iterative method* (BIM) (Papernot et al., 2018) and as *projected gradient descent* (PGD) (Madry et al., 2017). We refer to as PGD$_2$ a 2-norm version replacing (8) with

$$\mathbf{y}_{i+1} := \operatorname{proj}_{B_2[\mathbf{x};\epsilon]}(\mathbf{y}_i - \alpha\eta(\nabla_{\mathbf{x}} \ell(f(\mathbf{y}_i), t))), \tag{9}$$

where $\eta(\mathbf{x}) := \mathbf{x}/\|\mathbf{x}\|$ denotes 2-normalization, and projection is to the closed 2-norm ball $B_2[\mathbf{x}; \epsilon]$ of radius $\epsilon$ and center $\mathbf{x}$, followed again by element-wise clipping to $[0, 1]$. Although this method is part of Cleverhans library (Papernot et al., 2018), it is not published according to our knowledge.

**Target Success**. This family of attacks is typically more expensive. Szegedy et al. (2013) propose a Lagrangian formulation of problem (4)-(5), minimizing the cost function

$$J(\mathbf{y}, c) := \|\mathbf{x} - \mathbf{y}\|^2 + \lambda\ell(f(\mathbf{y}), t), \tag{10}$$

where variable $\lambda$ is a Lagrange multiplier for (5). They carry out this optimization by box-constrained L-BFGS.

The attack of Carlini & Wagner (2017), denoted by C&W in the sequel, pertains to this approach. A change of variable eliminates the box constraint, replacing $\mathbf{y} \in \mathcal{X}$ by $\sigma(\mathbf{w})$, where $\mathbf{w} \in \mathbb{R}^n$ and $\sigma$ is the element-wise sigmoid function. The classification loss encourages the logit $\log p_t$ to be less than any other $\log p_k$ for $k \neq t$ by at least margin $m \geq 0$,

$$\ell_m(\mathbf{p}, t) := [\log p_t - \max_{k \neq t} \log p_k + m]_+, \tag{11}$$

where $[\cdot]_+$ denotes the positive part. This function is similar to the multi-class SVM loss by Crammer and Singer (Crammer & Singer, 2001), where $m = 1$, and, apart from the margin, it is a hard version of negative cross-entropy $\ell$ where softmax is producing the classifier probabilities. It does not have the problem of being flat in the region of class $t$. The C&W attack uses the Adam optimizer (Kingma & Ba, 2015) to minimize the cost function

$$J(\mathbf{w}, \lambda) := \|\sigma(\mathbf{w}) - \mathbf{x}\|^2 + \lambda\ell_m(f(\sigma(\mathbf{w})), t). \tag{12}$$

for $\mathbf{w} \in \mathbb{R}^n$. When the margin is reached, loss $\ell_m$ vanishes and the distortion term pulls $\sigma(\mathbf{w})$ back towards $\mathbf{x}$, causing oscillations around the margin. This is repeated for different $\lambda$[2] by line search, which is expensive.

*Decoupling direction and norm* (DDN) (Rony et al., 2018) is iterating similarly to PGD$_2$ (9),

$$\mathbf{y}_{i+1} := \operatorname{proj}_{S[\mathbf{x};\rho_i]}(\mathbf{y}_i - \alpha\eta(\nabla_{\mathbf{x}} \ell(f(\mathbf{y}_i), t))), \tag{13}$$

---

[1]We define $\operatorname{proj}_A(\mathbf{u}) := \arg\min_{\mathbf{v} \in A} \|\mathbf{u} - \mathbf{v}\|$.
[2]Referred to as $c$ in Carlini & Wagner (2017).

but projection is to the sphere $S[\mathbf{x}; \rho_i]$ of radius $\rho_i$ and center $\mathbf{x}$, and the radius is adapted to the current distortion: It is set to $\rho_i = (1 - \gamma)\|\mathbf{y}_i - \mathbf{x}\|$ if $\mathbf{y}_i$ is adversarial and to $(1 + \gamma)\|\mathbf{y}_i - \mathbf{x}\|$ otherwise, where $\gamma \in (0, 1)$ is a parameter. Another major difference is that each iteration is concluded by a projection onto $\mathcal{X}$ (rather than $\hat{\mathcal{X}}$) by element-wise clipping to $[0, 1]$ and *rounding*.

**Discussion.** Optimizing around the class boundary is not a new idea. All of the above attacks do so in order to minimize distortion; implicitly, even attacks targeting distortion like PGD$_2$ do so, if the minimum parameter $\epsilon$ is sought (*cf*. Figure 1(a) and Section 4.2). Even *black-box* attacks do so (Brendel et al., 2018), without having access to the gradient function. The difference of our attack is that our updates are *along* the class boundary, *i.e.*, in a direction normal to the gradient. Deep-Fool (Moosavi-Dezfooli et al., 2016b) is a popular attack targeting success, that is not optimizing for distortion and not following a path around the class boundary.

### 2.3 OTHER RELATED WORK

**Optimization on manifolds.** In the context of deep learning, stochastic gradient descent on Riemanian manifolds has been studied, *e.g.* RSGD (Bonnabel, 2013) and RSVRG (Zhang et al., 2016). It is usually applied to manifolds whose geometry is known in analytic form, for instance Grassmann manifolds (Bonnabel, 2013), optimizing orthogonal matrices on Stiefel manifolds (Harandi & Fernando, 2016) or embedding trees on the Poincaré ball (Nickel & Kiela, 2017).

In most cases, the motivation is to optimize a very complex function (*e.g.* a classification loss) on a well-studied manifold, *e.g.* matrix manifold (Absil et al., 2009). On the contrary, we are optimizing a very simple quadratic function (the distortion) on a complex manifold not known in analytic form, *i.e.* a level set of the classification loss.

## 3 METHOD

Our attack is an iterative process with a fixed number $K$ of iterations. Stage 1 aims at quickly producing an adversarial image, whereas Stage 2 is a refinement phase decreasing distortion. The key property of our method is that while in the adversarial region during refinement, it tries to walk along the classification boundary by projecting the distortion gradient onto the tangent hyperplane of the boundary. Hence we call it *boundary projection* (BP).

### 3.1 STAGE 1

This stage begins at $\mathbf{y}_0 = \mathbf{x}$ and iteratively updates in the direction of the gradient of the loss function as summarized in Algorithm 1. The gradient is normalized and then scaled by two parameters: a fixed parameter $\alpha$ that is large s.t., with high probability, Stage 1 returns an adversarial image $\mathbf{y}_i \in \hat{\mathcal{X}}$; and a parameter $\gamma_i$ that is increasing linearly with iteration $i$ as follows

$$\gamma_i := \gamma_{\min} + \frac{i}{K + 1}(\gamma_{\max} - \gamma_{\min}), \tag{14}$$

such that updates are slow at the beginning to keep distortion low, then faster until the attack succeeds, where $\gamma_{\min} \in (0, \gamma_{\max})$ and $\gamma_{\max} = 1$. Clipping is element-wise.

---

**Algorithm 1** Stage 1

**Input:** $\mathbf{x}$: original image to be attacked
**Input:** $t$: true label (untargeted)
**Output:** $\mathbf{y}$ with $\pi(\mathbf{y}) \neq t$ or failure, iteration $i$
  1: Initialize $\mathbf{y}_0 \leftarrow \mathbf{x}$, $i \leftarrow 0$
  2: **while** $(\pi(\mathbf{y}_i) = t) \wedge (i < K)$ **do**
  3:     $\hat{\mathbf{g}} \leftarrow \eta(\nabla_{\mathbf{x}}\ell(f(\mathbf{y}_i), t))$
  4:     $\mathbf{y}_{i+1} \leftarrow \text{clip}_{[0,1]}(\mathbf{y}_i - \alpha\gamma_i\hat{\mathbf{g}})$
  5:     $i \leftarrow i + 1$
  6: **end while**

---

## 3.2 STAGE 2

Once Stage 1 has succeeded, Stage 2 continues by considering two cases: if $\mathbf{y}_i$ is adversarial, case OUT aims at minimizing distortion while staying in the adversarial region. Otherwise, case IN aims at decreasing the loss while controlling the distortion. Both work with a first order approximation of the loss around $\mathbf{y}_i$:

$$\ell(f(\mathbf{y}_i + \mathbf{u}), t) \approx \ell(f(\mathbf{y}_i), t) + \mathbf{u}^\top \mathbf{g}, \tag{15}$$

where $\mathbf{g} = \nabla_{\mathbf{x}} \ell(f(\mathbf{y}_i), t)$. The perturbation at iteration $i$ is $\boldsymbol{\delta}_i := \mathbf{y}_i - \mathbf{x}$. Stage 2 is summarized in Algorithm 2. Cases OUT and IN illustrated in Fig. 2 are explained below.

---

**Algorithm 2** Stage 2

**Input:** $t$: true label (untargeted), $i$ current iteration number
**Input:** $\mathbf{y}_i$: current adversarial image, $\epsilon$: target distortion
**Output:** $\mathbf{y}_K$

1: **while** $i < K$ **do**
2: $\quad \boldsymbol{\delta}_i \leftarrow \mathbf{y}_i - \mathbf{x}$ $\hfill \triangleright$ perturbation
3: $\quad \hat{\mathbf{g}} \leftarrow \eta(\nabla_{\mathbf{x}} \ell(f(\mathbf{y}_i), t))$ $\hfill \triangleright$ direction
4: $\quad r \leftarrow \langle \boldsymbol{\delta}_i, \hat{\mathbf{g}} \rangle$
5: $\quad$ **if** $\pi(\mathbf{y}_i) \neq t$ **then** $\hfill \triangleright$ OUT
6: $\quad\quad \epsilon \leftarrow \gamma_i \|\boldsymbol{\delta}_i\|$ $\hfill \triangleright$ target distortion
7: $\quad\quad \mathbf{v}^\star \leftarrow \mathbf{x} + r\hat{\mathbf{g}}$
8: $\quad\quad \mathbf{z} \leftarrow \mathbf{v}^\star + \eta(\mathbf{y}_i - \mathbf{v}^\star)\sqrt{[\epsilon^2 - r^2]_+}$
9: $\quad\quad \mathbf{y}_{i+1} \leftarrow Q_{\text{OUT}}(\mathbf{z}, \mathbf{y}_i)$
10: $\quad$ **else** $\hfill \triangleright$ IN
11: $\quad\quad \epsilon \leftarrow \|\boldsymbol{\delta}_i\| / \gamma_i$ $\hfill \triangleright$ target distortion
12: $\quad\quad \mathbf{z} \leftarrow \mathbf{y}_i - \left( r + \sqrt{\epsilon^2 - \|\boldsymbol{\delta}_i\|^2 + r^2} \right) \hat{\mathbf{g}}$
13: $\quad\quad \mathbf{y}_{i+1} \leftarrow Q_{\text{IN}}(\mathbf{z}, \mathbf{y}_i)$
14: $\quad$ **end if**
15: $\quad i \leftarrow i + 1$
16: **end while**

---

**Case OUT** takes as input $\mathbf{y}_i$ outside class $t$ region, *i.e.* $\pi(\mathbf{y}_i) \neq t$. We set a target distortion $\epsilon = \gamma_i \|\boldsymbol{\delta}_i\| < \|\boldsymbol{\delta}_i\|$ (14) such that updates decelerate to convergence once the attack has already succeeded. We then solve the following problem:

$$\mathbf{z} := \arg\min_{\mathbf{v} \in V} \|\mathbf{v} - \mathbf{y}_i\| \tag{16}$$

$$V := \arg\min_{\mathbf{v} \in P} |\|\mathbf{v} - \mathbf{x}\| - \epsilon|, \tag{17}$$

where $P := \{\mathbf{v} \in \mathbb{R}^n : \langle \mathbf{v} - \mathbf{y}_i, \hat{\mathbf{g}} \rangle = 0\}$ is the tangent hyperplane of the level set of the loss at $\mathbf{y}_i$, normal to $\hat{\mathbf{g}}$. The constraint $\mathbf{v} \in P$ aims at maintaining the value of the loss, up to the first order. On this hyperplane, $V$ is the set of points having distortion close to $\epsilon$.

Consider the projection $\mathbf{v}^\star := \mathbf{x} + r\hat{\mathbf{g}}$ of $\mathbf{x}$ onto that hyperplane, where $r := \langle \boldsymbol{\delta}_i, \hat{\mathbf{g}} \rangle$. If $r \geq \epsilon$, then $V = \{\mathbf{v}^\star\}$, and the solution of (16) is trivially $\mathbf{z} = \mathbf{v}^\star$. Note that $\mathbf{v}^\star = \mathbf{y}_i$ if $\boldsymbol{\delta}_i, \hat{\mathbf{g}}$ are collinear. If $r < \epsilon$, there is an infinity of solutions to (17). We pick the one closest to $\mathbf{y}_i$:

$$\mathbf{z} = \mathbf{v}^\star + \eta(\mathbf{y}_i - \mathbf{v}^\star)\sqrt{\epsilon^2 - r^2}. \tag{18}$$

This case is illustrated in Fig. 2(a), where $V$ is a circle that is the intersection of sphere $S[\mathbf{x}; \epsilon]$ and $P$; then $\mathbf{z}$ is the intersection of $V$ and the line through $\mathbf{y}_i$ and $\mathbf{v}^\star$.

Directly quantizing vector $\mathbf{z}$ onto $\mathcal{X}$ by $Q(\cdot)$, the component-wise rounding, modifies its norm (see App. A). This pulls down our effort to control the distortion. Instead, the process $Q_{\text{OUT}}(\mathbf{z}, \mathbf{y}_i)$ in line 9 looks for the scale $\beta$ of the perturbation to be applied s.t. $\|Q(\mathbf{y}_i + \beta(\mathbf{z} - \mathbf{y}_i))\| = \|\mathbf{z}\|$. This is done with a simple line search over $\beta$.

**Case IN** takes as input $\mathbf{y}_i$ inside class $t$ region, *i.e.* $\pi(\mathbf{y}_i) = t$. We set a target distortion $\epsilon = \|\boldsymbol{\delta}_i\| / \gamma_i > \|\boldsymbol{\delta}_i\|$ (14) such that updates decelerate as in Case OUT. We then solve the problem:

$$\mathbf{z} := \arg\min_{\mathbf{v} \in S[\mathbf{x}; \epsilon]} \langle \mathbf{v}, \hat{\mathbf{g}} \rangle, \tag{19}$$

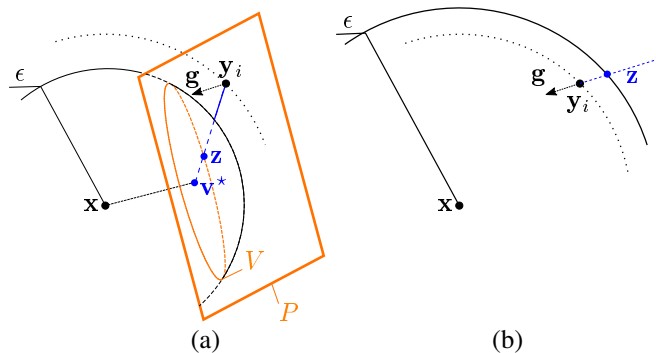

Figure 2: Refinement stage of BP. Case OUT when $|V| > 1$ (a); case IN (b). See text for details.

*i.e.*, find the point $\mathbf{z}$ at the intersection of sphere $S[\mathbf{x}; \epsilon]$ and the ray through $\mathbf{y}_i$ in the direction opposite of $\mathbf{g}$ as shown in Fig. 2(b). The solution is simple:

$$\mathbf{z} = \mathbf{y}_i - \left( r + \sqrt{\epsilon^2 - \|\boldsymbol{\delta}_i\|^2 + r^2} \right) \hat{\mathbf{g}}, \tag{20}$$

Vector $\mathbf{z}$ moves away from $\mathbf{y}_i$ along direction $-\hat{\mathbf{g}}$ by a step size so to reach $S[\mathbf{x}, \epsilon]$. Case IN is not guaranteed to succeed, but invoking it means that Stage 1 has succeeded.

Again a direct rounding jeopardizes the norm of the update $\mathbf{z} - \mathbf{y}_i$. Especially, quantization likely results in $Q(\mathbf{z}) = Q(\mathbf{y}_i)$ if $\|\mathbf{z} - \mathbf{y}_i\| < \beta_{\min} = 0.1$ (see App. A). Instead of a line search as in method OUT, line 13 just makes sure that this event will not happen: $Q_{\text{IN}}(\mathbf{z}, \mathbf{y}_i) = Q(\mathbf{y}_i + \beta(\mathbf{z} - \mathbf{y}_i))$ with $\beta = \max(1, \beta_{\min}/\|\mathbf{z} - \mathbf{y}_i\|)$.

## 4 EXPERIMENTS

In this section we compare our method *boundary projection* (BP) to the attacks presented in Sect. 2, namely: FGSM (Goodfellow et al., 2014), I-FGSM (Kurakin et al., 2016), PGD$_2$ (9), C&W (Carlini & Wagner, 2017), and DDN (Rony et al., 2018). This benchmark is carried out on three well-known datasets, with a different neural network for each.

### 4.1 DATASETS, NETWORKS, AND PARAMETERS

For the target distortion attacks *i.e.* FGSM, I-FGSM and PGD$_2$, we test a set of $\epsilon$ and calculate $P_{\text{suc}}$ and $\overline{D}$ according to our evaluation protocol (*cf*. section 4.2). For C&W, we test several parameter settings and pick up the optimum setting as specified below. For DDN, the parameter settings are the default (Rony et al., 2018), *i.e.* $\epsilon_0 = 1.0$ and $\gamma = 0.05$. Below we specify different networks and parameters for each dataset.

**MNIST (LeCun et al., 2010).** We use is a simple network with three convolutional layers and one fully connected layer achieving accuracy $0.99$, referred to as C4. The first convolutional layer has $64$ features, kernel of size $8$ and stride $2$; the second has $128$ features, kernel $6$ and stride $2$; the third has also $128$ features, but kernel $5$ and stride $1$. It uses LeakyRelu activation (Maas et al., 2013).

*Parameters*. We set $\alpha = 0.08$ for I-FGSM and $\alpha = \epsilon/2$ for PGD$_2$. For C&W: for $5 \times 20$ iterations[3], learning rate $\eta = 0.5$ and initial constant $\lambda = 1.0$; for $1 \times 100$ iterations, $\eta = 0.1$ and $\lambda = 10.0$.

**CIFAR10 (Krizhevsky & Hinton, 2009).** We use a simple CNN network with nine convolutional layers, two max-pooling layers, ending in global average pooling and a fully connected layer. Its accuracy is $0.927$. Batch normalization (Ioffe & Szegedy, 2015) is applied after every convolutional layer. It also uses LeakyRelu.

*Parameters*. We set $\alpha = 0.08$ for I-FGSM and $\alpha = \epsilon/2$ for PGD$_2$. For C&W: for $5 \times 20$ iterations, learning rate $\eta = 0.1$ and initial constant $\lambda = 0.1$; for $1 \times 100$ iterations, $\eta = 0.01$, and $\lambda = 1.0$.

---

[3]C&W performs line search on $\lambda$: "$5 \times 20$" means 5 values of $\lambda$, 20 iterations for each.

|  | # Grads | $P_{\text{suc}}$ | $\overline{D}$ |
|---|---|---|---|
| Rounding in the end | 20 | 1.00 | 1.44 |
|  | 100 | 1.00 | 1.43 |
| Rounding at each iteration | 20 | 1.00 | 0.41 |
|  | 100 | 1.00 | 0.32 |
| Rounding with $Q_{\text{IN}}, Q_{\text{OUT}}$ | 20 | 1.00 | 0.35 |
|  | 100 | 1.00 | 0.28 |

Table 1: Success probability $P_{\text{suc}}$ and average distortion $\overline{D}$ of our method BP on ImageNet with different *quantization strategies*.

**ImageNet (Kurakin et al., 2018)** comprises 1,000 images from ImageNet (Deng et al., 2009). We use InceptionV3 (Szegedy et al., 2016) whose accuracy is 0.96.

*Parameters.* We set $\alpha = 0.08$ for I-FGSM and $\alpha = 3$ for PGD$_2$. For C&W: for $5 \times 20$ iterations, learning rate $\eta = 0.01$ and initial constant $\lambda = 20$; for $1 \times 100$ iterations, $\eta = 0.01$ and $\lambda = 1.0$.

## 4.2 EVALUATION PROTOCOL

We evaluate an attack by its runtime, two global statistics $P_{\text{suc}}$ and $\overline{D}$, and by an operating characteristic curve $D \to \mathsf{P}(D)$ measuring distortion *vs.* probability of success as described below.

Since we focus on the *speed-distortion trade-off*, we measure the required time for all attacks. For the iterative attacks, the complexity of one iteration is largely dominated by the computation of the gradient, which requires one forward and one backward pass through the network. It is thus fair to gauge their complexity by this number, referred to as *iterations* or '# Grads'. Indeed, the actual timings of 100 iterations for I-FGSM, PGD$_2$, C&W, DDN and BP are 1.08, 1.36, 1.53, 1.46 and 1.17 s/image on average respectively on ImageNet, using Tensorflow, Cleverhans implementation for I-FGSM and C&W, and authors implementation for DDN.

We measure distortion when *the adversarial images are quantized* by rounding each element to the nearest element in $\mathcal{X}$. This makes sense since adversarial images are meant to be stored or communicated as images rather than real-valued matrices. DDN and BP adversarial images are already quantized. For reference, we report distortion without quantization in Appendix B.3.

Given a test set of $N'$ images, we only consider its subset $X$ of $N$ images that are classified correctly without attack. The accuracy of the classifier is $N/N'$. Let $X_{\text{suc}}$ be the subset of $X$ with $N_{\text{suc}} := |X_{\text{suc}}|$ where the attack succeeds and let $D(\mathbf{x}) := \|\mathbf{x} - \mathbf{y}\|$ be the distortion for image $\mathbf{x} \in X_{\text{suc}}$. The global statistics are the *success probability* $P_{\text{suc}}$ and *conditional average distortion* $\overline{D}$

$$P_{\text{suc}} := \frac{N_{\text{suc}}}{N}, \quad \overline{D} := \frac{1}{N_{\text{suc}}} \sum_{\mathbf{x} \in X_{\text{suc}}} D(\mathbf{x}). \tag{21}$$

Here, $\overline{D}$ is conditioned on success. Indeed, distortion makes no sense for a failure.

We define the *operating characteristic* of a given attack over the set $X$ as the function $\mathsf{P} : [0, D_{\max}] \to [0, 1]$, where $D_{\max} := \max_{\mathbf{x} \in X_{\text{suc}}} D(\mathbf{x})$. Given $D \in [0, D_{\max}]$, $\mathsf{P}(D)$ is the probability of success subject to distortion being upper bounded by $D$,

$$\mathsf{P}(D) := \frac{1}{N} |\{\mathbf{x} \in X_{\text{suc}} : D(\mathbf{x}) \le D\}|. \tag{22}$$

This function increases from $\mathsf{P}(0) = 0$ to $\mathsf{P}(D_{\max}) = P_{\text{suc}}$. We sample one intermediate point: $P_{\text{upp}} := \mathsf{P}(D_{\text{upp}})$ is the success rate within a distortion upper bounded by $D_{\text{upp}} \in (0, D_{\max})$.

It is difficult to define a fair comparison of *distortion targeting* attacks to *success targeting* attacks (see section 2.2). For the first family, we run a given attack several times over the test set with different target distortion $\epsilon$. The attack succeeds on image $\mathbf{x} \in X$ if it succeeds on at least one of the runs, and the distortion $D(\mathbf{x})$ is the minimum distortion over all successful runs. All statistics are then evaluated as above.

| Attack | # Grads | MNIST | | | CIFAR10 | | | ImageNet | | |
|---|---|---|---|---|---|---|---|---|---|---|
| | | $P_{\text{suc}}$ | $\overline{D}$ | $P_{\text{upp}}$ | $P_{\text{suc}}$ | $\overline{D}$ | $P_{\text{upp}}$ | $P_{\text{suc}}$ | $\overline{D}$ | $P_{\text{upp}}$ |
| FGSM | 1 | 0.99 | 5.80 | 0.00 | 0.95 | 5.65 | 0.00 | 0.88 | 9.18 | 0.00 |
| I-FGSM | 20 | 1.00 | 3.29 | 0.17 | 1.00 | 3.54 | 0.00 | 1.00 | 4.90 | 0.00 |
| | 100 | 1.00 | 3.23 | 0.18 | 1.00 | 3.53 | 0.00 | 1.00 | 4.90 | 0.00 |
| $PGD_2$ | 20 | 1.00 | 1.80 | 0.63 | 1.00 | 0.66 | 0.76 | 0.63 | 3.63 | 0.00 |
| | 100 | 1.00 | 1.74 | 0.66 | 1.00 | 0.60 | 0.84 | 1.00 | 1.85 | 0.00 |
| C&W | $5\times20$ | 1.00 | 1.94 | 0.56 | 0.99 | 0.56 | 0.81 | 1.00 | 1.70 | 0.00 |
| | $1\times100$ | 0.98 | 1.90 | 0.57 | 0.87 | 0.38 | 0.76 | 0.97 | 2.57 | 0.00 |
| DDN | 20 | 0.82 | 1.40 | 0.70 | 1.00 | 0.63 | 0.74 | 0.99 | 1.18 | 0.05 |
| | 100 | 1.00 | 1.41 | 0.87 | 1.00 | 0.21 | 0.98 | 1.00 | 0.43 | 0.97 |
| BP (this work) | 20 | 1.00 | 1.45 | 0.86 | 0.97 | 0.49 | 0.87 | 1.00 | 0.35 | 0.96 |
| | 100 | 1.00 | 1.37 | 0.91 | 0.97 | 0.30 | 0.97 | 1.00 | 0.28 | 1.00 |

Table 2: Success probability $P_{\text{suc}}$ and average distortion $\overline{D}$ with quantization. $P_{\text{upp}}$ is the success rate under distortion budget $D_{\text{upp}} = 2$ for MNIST, $0.7$ for CIFAR10, and $1$ for ImageNet.

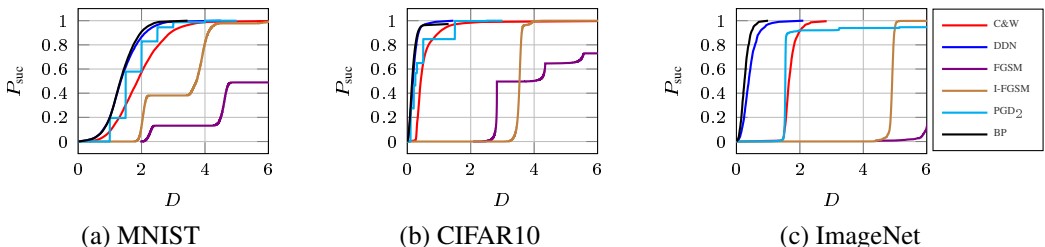

(a) MNIST  (b) CIFAR10  (c) ImageNet

Figure 3: Operating characteristics on MNIST, CIFAR10 and ImageNet. The number of iterations is $5 \times 20$ for C&W and $100$ for I-FGSM, $PGD_2$, DDN and our BP.

## 4.3 QUANTIZATION

Before addressing the benchmark, Table 1 shows the critical role of quantization in our method BP. Since this attack is iterative and works with continuous vectors, one may quantize only at the end of the process, or at the end of each iteration. Another option is to anticipate the detrimental action of quantizing by adapting the length of each step accordingly, as done by $Q_{\text{IN}}(\cdot)$ and $Q_{\text{OUT}}(\cdot)$ in Algorithm 2. The experimental results show that the key is to quantize often so to let the next iterations compensate. Anticipating and adapting gives a substantial extra improvement.

## 4.4 ATTACK EVALUATION

Table 2 summarizes the global statistics of the benchmark. Fig. 3 offers a more detailed view per dataset with operating characteristic plots.

In terms of average distortion, all iterative attacks perform much better than the single-step FGSM. The performances of C&W are on par with those of I-FGSM, which is unexpected for this more elaborated attack design. The reason is that C&W is put under stress in our benchmark. It usually requires a bigger number of iterations to deliver high quality images. Note that it is possible to avoid the line search on parameter $\lambda$ as shown in row $1 \times 100$. However, it requires a fine tuning so that this single value works over all the images of the dataset. This is not possible for ImageNet.

DDN and our method BP are clearly ahead of the benchmark. DDN yields lower distortion on MNIST at fewer iterations, but its probability of success is not satisfying. DDN is indeed better than BP only on CIFAR10 at 100 iterations. Fig. 3 reveals that the two attacks have similar operating characteristic on all datasets but this is because it refers to 100 iterations.

In terms of success rate, FGSM fails on MNIST; on CIFAR10, I-FGSM and $PGD_2$ fail as well; finally on ImageNet, C&W fails too. DDN also fails on ImageNet at 20 iterations.

Increasing the number of iterations helps but not at the same rate for all the attacks. For instance, going from 20 to 100 iterations is waste of time for I-FGSM while it is essential for decreasing the distortion of DDN or making $PGD_2$ efficient on ImageNet. Most importantly, our attack BP brings a dramatic improvement in the speed *vs*. distortion trade-off. Just within 20 iterations, the

| | | MNIST | | | | | | CIFAR10 | | | | | |
|---|---|---|---|---|---|---|---|---|---|---|---|---|---|
| Attack → | | PGD$_2$ | | DDN | | BP | | PGD$_2$ | | DDN | | BP | |
| ↓ Defense | | 20 | 100 | 20 | 100 | 20 | 100 | 20 | 100 | 20 | 100 | 20 | 100 |
| baseline | $P_{\text{suc}}$ | 1.00 | 1.00 | 0.82 | 1.00 | 1.00 | 1.00 | 1.00 | 1.00 | 1.00 | 1.00 | 0.97 | 0.97 |
| | $\overline{D}$ | 1.80 | 1.74 | 1.40 | 1.41 | 1.45 | 1.37 | 0.66 | 0.59 | 0.63 | 0.21 | 0.49 | 0.30 |
| | $P_{\text{upp}}$ | 0.63 | 0.66 | 0.70 | 0.87 | 0.86 | 0.91 | 0.76 | 0.84 | 0.74 | 0.98 | 0.87 | 0.97 |
| FGSM | $P_{\text{suc}}$ | 1.00 | 1.00 | 0.51 | 1.00 | 0.89 | 1.00 | 1.00 | 1.00 | 1.00 | 1.00 | 0.99 | 1.00 |
| | $\overline{D}$ | 1.92 | 1.85 | 1.28 | 1.60 | 1.92 | 1.58 | 0.68 | 0.62 | 0.59 | 0.24 | 0.67 | 0.24 |
| | $P_{\text{upp}}$ | 0.48 | 0.53 | 0.44 | 0.72 | 0.53 | 0.73 | 0.72 | 0.79 | 0.80 | 0.98 | 0.72 | 0.99 |
| DDN | $P_{\text{suc}}$ | 0.99 | 1.00 | 0.29 | 1.00 | 0.99 | 1.00 | 1.00 | 1.00 | 0.98 | 1.00 | 1.00 | 1.00 |
| | $\overline{D}$ | 3.03 | 2.89 | 1.68 | 2.38 | 2.69 | 2.27 | 0.95 | 0.94 | 0.77 | 0.71 | 0.75 | 0.68 |
| | $P_{\text{upp}}$ | 0.12 | 0.14 | 0.20 | 0.32 | 0.28 | 0.34 | 0.52 | 0.52 | 0.54 | 0.55 | 0.56 | 0.58 |
| BP | $P_{\text{suc}}$ | 0.94 | 0.96 | 0.36 | 1.00 | 0.95 | 1.00 | 1.00 | 1.00 | 0.97 | 1.00 | 1.00 | 1.00 |
| | $\overline{D}$ | 3.14 | 3.12 | 1.65 | 2.81 | 2.98 | 2.73 | 0.96 | 0.94 | 0.75 | 0.70 | 0.76 | 0.69 |
| | $P_{\text{upp}}$ | 0.15 | 0.15 | 0.24 | 0.27 | 0.25 | 0.26 | 0.55 | 0.55 | 0.57 | 0.59 | 0.56 | 0.59 |

Table 3: Success probability $P_{\text{suc}}$, average distortion $\overline{D}$, and success rate $P_{\text{upp}}$ under *adversarial training* defense with I-FGSM, DDN, or BP (with 20 iterations) as the reference attack. For the MNIST, the model is trained from scratch. For CIFAR10, it is fine-tuned for 30 extra epochs as suggested by Rony et al. (2018) with DDN and BP, and trained from scratch for 200 epochs with FGSM. $P_{\text{upp}}$ measured at distortion $D_{\text{upp}} = 2$ for MNIST, and $0.7$ for CIFAR10.

distortion achieved on ImageNet is very low compared to the others. Appendix B.2 shows the speed *vs*. distortion trade-off in more detail.

Statistics of BP stages are as follows: On CIFAR-10 and MNIST, Stage 1 takes 7 iterations on average. On ImageNet, Stage 1 takes on average 3 iterations out of 20, or 8 iterations out of 100.

Appendix C shows examples of images along with corresponding adversarial examples and perturbations for different methods.

### 4.5 Defense evaluation with adversarial training

We also test under adversarial training (Goodfellow et al., 2014). The network is re-trained with a dataset composed of the original training set and the corresponding adversarial images. This training is special: at the end of each epoch, the network is updated and fixed, then the adversarial images for this new update are forged by some reference attack, and the next epoch starts with this new set. This is tractable only if the reference attack is fast. We use it with FGSM as the reference attack.

It is more interesting to study DDN and BP as alternatives to FGSM: at 20 iterations, they are fast enough to play the role of the reference attack in adversarial training. In this case, we follow the training process suggested by Rony et al. (2018): the model is first trained on clean examples, then fine-tuned for 30 iterations with adversarial examples. As shown in Table 3, DDN and BP perform equally better than FGSM on CIFAR10, in terms of either average distortion or success rate. Among the reliable attacks (*i.e.* whose $P_{\text{suc}}$ is close to 1), the worst attack now requires a distortion three times larger than the distortion of the worst attack without defense. In the same way, on MNIST, the distortion of the worst case attack doubles going from $1.37$ (baseline) to $2.73$ (BP defense). In most cases, BP is a better defense than DDN, forcing the attacker to have 20% more distortion. Note that for a given defense, the strongest attack is almost always BP.

## 5 Discussion

The main idea of BP is to travel on the manifold defined by the class boundary while seeking to minimize distortion. This travel is operated by the refinement stage, which alternates on both sides of the boundary, but attempts to stay mostly in the adversarial region. Referring to section 2.1, BP is in effect doing for the *target success* problem what PGD$_2$ is doing for the *target distortion* problem: BP minimizes distortion on the class boundary manifold (a level set of the classification loss), while PGD$_2$ minimizes the classification loss on a sphere (a level set of the distortion).

BP also takes into account the detrimental effect of *quantization*. By doing so, the amplitude of the perturbation is controlled from one iteration to another. The main advantage of our attack is the small number of iterations required to achieve both reliability (probability of success close to one) and high quality (low average distortion).

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

## A    PREDICTING DISTORTION AFTER QUANTIZATION

This appendix aims at predicting the norm of the update after quantization, assuming that it is independent from the computation of the perturbation. Iteration $i$ starts with a quantized image $\mathbf{y}_i \in \mathcal{X}$, adds update $\mathbf{u} \in \mathbb{R}^n$, and then quantizes s.t. $\mathbf{y}_{i+1} = Q(\mathbf{y}_i + \mathbf{u})$. Quantization is done by rounding with $\Delta := 1/(L-1)$ the quantization step. Pixel $j$ is quantized to

$$y_{i+1,j} = y_{i,j} + e_j \tag{23}$$

for some $e_j \in \Delta\mathbb{Z}$ such that $u_j \in (e_j - \Delta/2, e_j + \Delta/2]$. Border effects where $y_{i,j} + e_j \notin \mathcal{X}$ are neglected.

We now take a statistical point of view where the update is modelled by a random vector $\mathbf{U}$ uniformly distributed over the hypersphere of radius $\rho$. That parameter $\rho$ is the norm of the perturbation before quantization. This yields random quantization values, denoted by $E_j \in \Delta\mathbb{Z}$ for pixel $j$. The distortion between the two images is

$$D^2 = \sum_{j=1}^{n} (y_{i+1,j} - y_{i,j})^2 = \sum_{j=1}^{n} E_j^2. \tag{24}$$

A common approach in source coding theory is the additive noise model for quantization error in the high resolution regime (Gersho & Gray, 1991). It states that $E_j = U_j + Q_j$ where $Q_j \in (-\Delta/2, \Delta/2]$ is the quantization error. When $\rho \gg \Delta$, then $Q_j$ becomes uniformly distributed (s.t. $\mathrm{E}(Q_j) = 0$ and $\mathrm{E}(Q_j^2) = \Delta^2/12$) and independent of $U_j$ (s.t. $\mathrm{E}(U_j Q_j) = \mathrm{E}(U_j)\mathrm{E}(Q_j) = 0$). Under these assumptions, Eq. 24 simplifies in expectation to:

$$\mathrm{E}(D^2) = \mathrm{E}\left( \sum_{j=1}^{n} U_j^2 + Q_j^2 + 2U_j Q_j \right) = \rho^2 + n\frac{\Delta^2}{12}. \tag{25}$$

This shows that quantization increases the distortion on expectation.

Yet, this simple analysis is wrong outside the high resolution regime, and we need to be more careful. The expectation of a sum is always the sum of the expectations, whatever the dependence between the summands: $\mathrm{E}(D^2) = \sum_{j=1}^{n} \mathrm{E}(E_j^2) = n\mathrm{E}(E_j^2)$ with

$$\mathrm{E}(E_j^2) = \Delta^2 \sum_{\ell=0}^{L-1} \ell^2 \mathrm{P}(|E_j| = \ell\Delta). \tag{26}$$

We need the distribution of $E_j$ to compute the expected distortion after quantizarion. This random variable $E_j$ takes a value depending on the scalar product $S_j := \mathbf{U}^\top \mathbf{c}_j$, where $\mathbf{c}_j$ is the $j$-th canonical vector. This scalar product lies in $[-\rho, \rho]$, so that $\mathrm{P}(E_j \geq \ell\Delta) = 0$ if $\ell\Delta - \Delta/2 > \|\rho\|$. Otherwise, $E_j \geq \ell\Delta$ when $|S_j| \geq \ell\Delta - \Delta/2$, which happens when $\mathbf{U}$ lies inside the dual hypercone of axis $\mathbf{c}_j$ and semi-angle $\theta(\ell) = \arccos(s(\ell))$ with $s(\ell) := (2\ell - 1)\Delta/2\|\rho\|$. The probability of this event is equal to the ratio of the solid angles of this dual hypercone and the full space $\mathbb{R}^n$. This quantity can be expressed via the incomplete regularized beta function $I$, and approximately equals $2\Phi(\sqrt{n}s(\ell)/2)$ for large $n$. In the end, $\forall \ell \in \{0, \ldots, L-1\}$,

$$\mathrm{P}(|E_j| \geq \ell\Delta) = \begin{cases} 1, & \text{if } \ell = 0 \\ 1 - I_{s(\ell)^2}(1/2, (n-1)/2), & \text{if } 0 \leq s(\ell) \leq 1 \\ 0, & \text{otherwise} \end{cases}$$

Computing $\mathrm{E}(D^2)$ is now possible because $\mathrm{P}(|E_j| = \ell\Delta) = \mathrm{P}(|E_j| \geq \ell\Delta) - \mathrm{P}(|E_j| \geq (\ell+1)\Delta)$. This expected distortion after quantization depends on $\Delta$, $n$, and $\rho$ the norm of the perturbation before quantization. Figure 4 shows that quantization reduces the distortion outside the high resolution regime. Indeed, $\sqrt{\mathrm{E}(D^2)}$ is close to 0 for $\rho < 0.1$ when $n = 3 * 299^2$ (*i.e.* ImageNet). When the update has a small norm $\rho$, quantization is likely to kill it, $\mathbf{y}_{i+1} = \mathbf{y}_i$, and we waste one iteration. On the contrary, $\sqrt{\mathrm{E}(D^2)}$ converges to $\sqrt{\rho^2 + n\Delta^2/12}$ for large $\rho$ (*i.e.* in the high resolution regime). Note that the ratio of the distortions before and after quantization $\rho/\sqrt{\mathrm{E}(D^2)}$ quickly converges to 1 for large $\rho$.

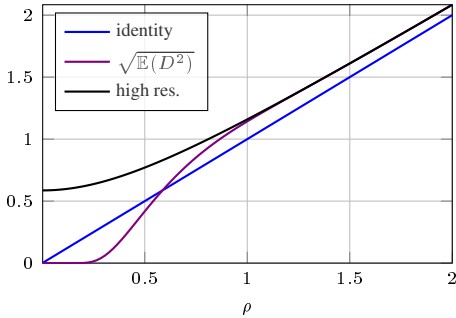

Figure 4: $\sqrt{\mathrm{E}(D^2)}$ as a function of $\rho$ for $n = 3 * 299^2$ and $\Delta = 1/255$.

# B ADDITIONAL EXPERIMENTS

## B.1 PARAMETER STUDY

There are two parameters in BP: $\alpha$ and $\gamma_{min}$. Both determine the step size of stage 1, while $\gamma_{min}$ also determines the step size of stage 2. We consider 4 values for $\alpha$, *i.e.* $1, 2, 3, 4$ and 9 values for $\gamma_{min}$, *i.e.* $0.1, 0.2, ..., 0.9$. For each pair of values, we evaluate BP with 20 iterations on a validation set, which we define as a random subset sampled of the training set: 10000 images for MNIST and CIFAR10, and 1000 images for ImageNet. As shown in Fig. 5, success probability is close to one in all cases, while average distortion is in general stable up to $\gamma_{min} = 0.8$. We choose $\alpha = 2$ and $\gamma_{min} = 0.7$ for all experiments.

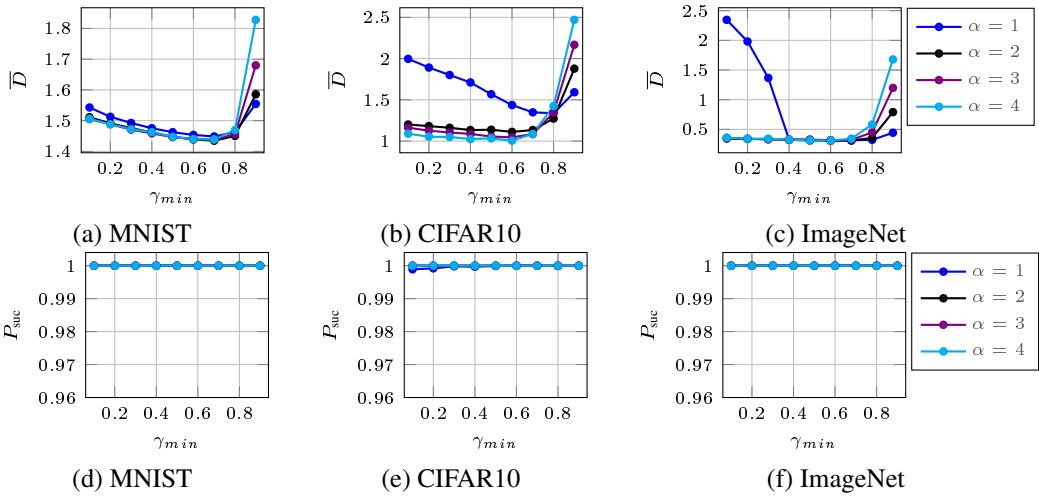

Figure 5: Success probability $P_{\text{suc}}$ and average distortion $\overline{D}$ for different values of parameters $\alpha$ and $\gamma_{min}$ of BP with 20 iterations.

## B.2 SPEED *vs*. DISTORTION TRADE-OFF

Figure 6(a) is a graphical view of some results reported in Table 2 with more choices of number of iterations between 20 and 100, and only for ImageNet where our performance gain is the most significant. Just within 20 iterations, its distortion $\overline{D}$ is already so much lower than that of other attacks, that its decrease (-20% at 100 iterations) is not visible in Fig. 6. On the contrary, more iterations are useless for I-FGSM, and PGD$_2$ can achieve low distortion only with a number of iterations bigger than 50. Figure 6(b) confirms that the probability of success is close to 1 for both DDN and BP for the numbers of iterations considered.

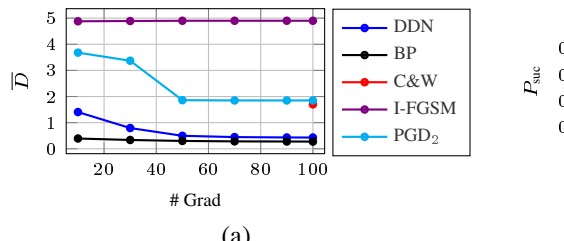 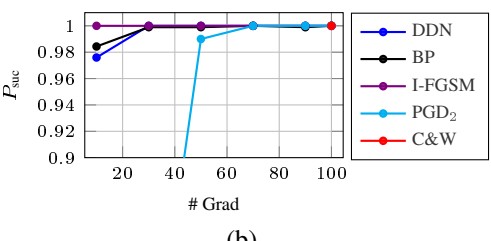

(a)                                                     (b)

Figure 6: (a) Average distortion *vs.* number of iterations for I-FGSM, $PGD_2$, C&W, DDN and our method BP on ImageNet. I-FGSM is not improving with iterations because it is constrained by $\epsilon$. (b) Corresponding probability of success *vs.* number of iterations for $PGD_2$ and BP.

## B.3   Attack evaluation without quantization

Table 4 is the equivalent of Table 2 but without the integral constraint: the attack is free to output any real matrix provided that the pixel values all belong to $[0, 1]$. When the distortion is large, there is almost no difference. The model of App. A explains this: We are in the high resolution regime and the extra term in Eq. 25 is negligible compared to the perturbation distortion before quantization. This is especially true when the number of samples $n$ is small (*i.e.* MNIST, and to some extend, CIFAR-10).

When an attack delivers low distortion on average with real matrices, the quantization may lower the probability of success. This is especially true with the iterative attacks finding adversarial examples just nearby the border between the two classes. Quantization jeopardizes this point and sometimes brings it back in the true class region. More importantly, the impact of the quantization on the distortion is no longer negligible. This is clearly visible when comparing Table 4 and Table 2 for DDN and BP over ImageNet.

Similarly, Fig. 7 is the equivalent of Fig. 3 without the integral constraint. By comparing the two figures, it can be seen that $PGD_2$ and C&W, but also DDN and BP, are improving on ImageNet by having significantly lower distortion. This agrees with measurements of success rate in Table 4, where $PGD_2$ and C&W are not failing as they do in Table 2 with quantization. Our BP is still the strongest attack over all datasets.

| Attack | # Grads | MNIST | | | CIFAR10 | | | ImageNet | | |
|---|---|---|---|---|---|---|---|---|---|---|
| | | $P_{\text{suc}}$ | $\overline{D}$ | $P_{\text{upp}}$ | $P_{\text{suc}}$ | $\overline{D}$ | $P_{\text{upp}}$ | $P_{\text{suc}}$ | $\overline{D}$ | $P_{\text{upp}}$ |
| FGSM | 1 | 0.99 | 5.81 | 0.00 | 0.97 | 4.78 | 0.00 | 0.85 | 3.02 | 0.00 |
| I-FGSM | 20 | 1.00 | 3.22 | 0.27 | 1.00 | 3.54 | 0.00 | 1.00 | 4.47 | 0.00 |
| | 100 | 1.00 | 3.16 | 0.29 | 1.00 | 3.53 | 0.00 | 1.00 | 4.47 | 0.00 |
| $PGD_2$ | 20 | 1.00 | 1.76 | 0.63 | 1.00 | 0.51 | 0.77 | 0.64 | 3.94 | 0.36 |
| | 100 | 1.00 | 1.70 | 0.66 | 1.00 | 0.43 | 0.85 | 0.95 | 1.11 | 0.61 |
| C&W | $5 \times 20$ | 1.00 | 1.93 | 0.56 | 1.00 | 0.56 | 0.81 | 1.00 | 1.37 | 0.23 |
| | $1 \times 100$ | 1.00 | 1.89 | 0.57 | 0.97 | 0.38 | 0.84 | 1.00 | 1.87 | 0.06 |
| DDN | 20 | 0.82 | 1.39 | 0.70 | 1.00 | 0.62 | 0.74 | 1.00 | 0.76 | 0.95 |
| | 100 | 1.00 | 1.41 | 0.87 | 1.00 | 0.20 | 0.98 | 1.00 | 0.28 | 0.99 |
| BP (this work) | 20 | 1.00 | 1.41 | 0.86 | 0.97 | 0.33 | 0.87 | 1.00 | 0.20 | 1.00 |
| | 100 | 1.00 | 1.35 | 0.91 | 0.97 | 0.18 | 0.97 | 1.00 | 0.16 | 1.00 |

Table 4: Success probability $P_{\text{suc}}$ and average distortion $\overline{D}$ *without quantization*. $P_{\text{upp}}$ measured at $D_{\text{upp}} = 2$ for MNIST, $0.7$ for CIFAR10, and $1$ for ImageNet.

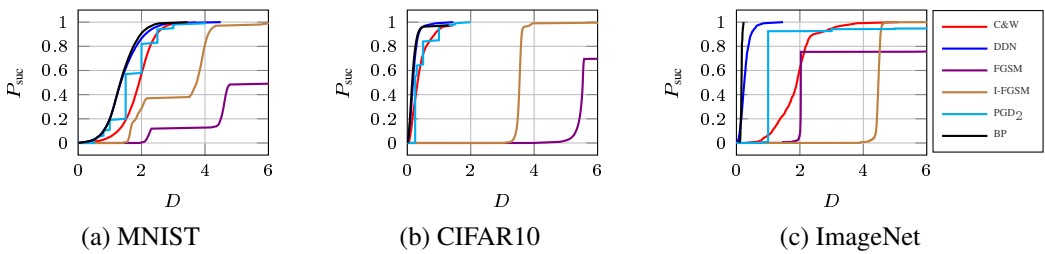

Figure 7: Operating characteristics on MNIST, CIFAR10 and ImageNet *without quantization*. The number of iterations is $5 \times 20$ for C&W and $100$ for I-FGSM, PGD$_2$, DDN and our BP.

## B.4 ATTACK EVALUATION ON ROBUST MODELS

Table 5 is similar to Table 2 but is evaluating attacks on robust models. In particular, on MNIST and CIFAR10, we use the same models as described in Section 4.1, which we adversarially train according to Madry et al. (2017). On ImageNet, we use off-the shelf[4] InceptionV3 obtained by *ensemble adversarial training* on four models Tramèr et al. (2017a).

In general, DDN and BP outperform all other attacks in terms of either average distortion $\overline{D}$ or success rate $P_{upp}$. On ImageNet in particular, all other attacks have significantly higher distortion and fail in terms of success rate. DDN and BP have similar performance on CIFAR10. On MNIST, DDN fails in terms of probability of success at 20 iterations, while at 100 iterations BP is superior. On ImageNet, DDN has significantly greater distortion than BP and fails in terms of success rate at 20 iterations, while at 100 iterations BP still has lower distortion.

Fig. 8 is showing a more detailed view of operating characteristics, similarly to Fig. 3 for models trained on natural images. We can see that BP is still ahead of the competition. It is close to DDN, but this is because Fig. 8 refers to 100 iterations. The two attacks outperform all others by a large margin.

| Attack | # Grads | MNIST (Madry et al., 2017) | | | CIFAR10 (Madry et al., 2017) | | | ImageNet (Tramèr et al., 2017a) | | |
|---|---|---|---|---|---|---|---|---|---|---|
| | | $P_{suc}$ | $\overline{D}$ | $P_{upp}$ | $P_{suc}$ | $\overline{D}$ | $P_{upp}$ | $P_{suc}$ | $\overline{D}$ | $P_{upp}$ |
| FGSM | 1 | 0.48 | 5.69 | 0.05 | 0.98 | 6.21 | 0.00 | 0.44 | 2.98 | 0.00 |
| I-FGSM | 20 | 1.00 | 4.99 | 0.08 | 1.00 | 4.53 | 0.00 | 1.00 | 4.92 | 0.00 |
| | 100 | 1.00 | 4.99 | 0.08 | 1.00 | 4.56 | 0.00 | 1.00 | 4.93 | 0.00 |
| PGD$_2$ | 20 | 0.99 | 2.76 | 0.19 | 1.00 | 1.03 | 0.41 | 0.76 | 2.14 | 0.00 |
| | 100 | 1.00 | 2.68 | 0.20 | 1.00 | 1.02 | 0.41 | 0.98 | 1.59 | 0.00 |
| C&W | $5 \times 20$ | 0.99 | 2.75 | 0.27 | 0.98 | 1.41 | 0.22 | 0.98 | 2.85 | 0.00 |
| | $1 \times 100$ | 0.94 | 2.22 | 0.34 | 0.60 | 0.77 | 0.27 | 0.97 | 2.41 | 0.00 |
| DDN | 20 | 0.43 | 1.61 | 0.32 | 0.97 | 0.92 | 0.41 | 0.99 | 1.10 | 0.23 |
| | 100 | 1.00 | 2.12 | 0.48 | 1.00 | 0.87 | 0.42 | 1.00 | 0.34 | 0.98 |
| BP (this work) | 20 | 1.00 | 2.17 | 0.46 | 1.00 | 0.94 | 0.41 | 1.00 | 0.35 | 0.94 |
| | 100 | 1.00 | 2.00 | 0.51 | 1.00 | 0.88 | 0.43 | 1.00 | 0.23 | 0.99 |

Table 5: Success probability $P_{suc}$, average distortion $\overline{D}$, and success rate $P_{upp}$ under *adversarial training* with PGD as the reference attack, following (Madry et al., 2017) for MNIST and CIFAR10; and *ensemble adversarial training* (Tramèr et al., 2017a) for ImageNet. $P_{upp}$ measured at $D_{upp} = 2$ for MNIST, $0.7$ for CIFAR10, and $1$ for ImageNet.

---

[4]`https://github.com/tensorflow/models/tree/master/research/adv_`
`imagenet_models`

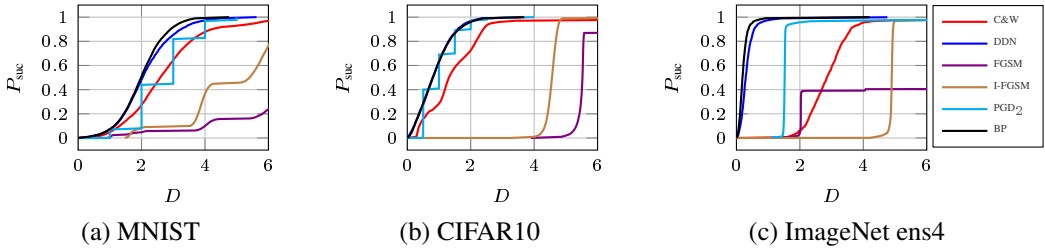

(a) MNIST        (b) CIFAR10        (c) ImageNet ens4

Figure 8: Operating characteristics of attacks against *robust models*: *adversarial training* with PGD as the reference attack (Madry et al., 2017) for MNIST and CIFAR10, and *ensemble* adversarial training (Tramèr et al., 2017a) for ImageNet. The number of iterations is $5 \times 20$ for C&W and $100$ for I-FGSM, $PGD_2$, DDN and our BP.

## C  ADVERSARIAL IMAGE EXAMPLES

Fig. 9 shows the worst-case ImageNet example for BP along with the adversarial examples generated by all methods and the corresponding normalized perturbations. FGSM has the highest distortion over all methods in this example and BP the lowest. DDN has the highest $\infty$-norm distortion. Observe that for no method is the perturbation visible, although this is a worst-case example.

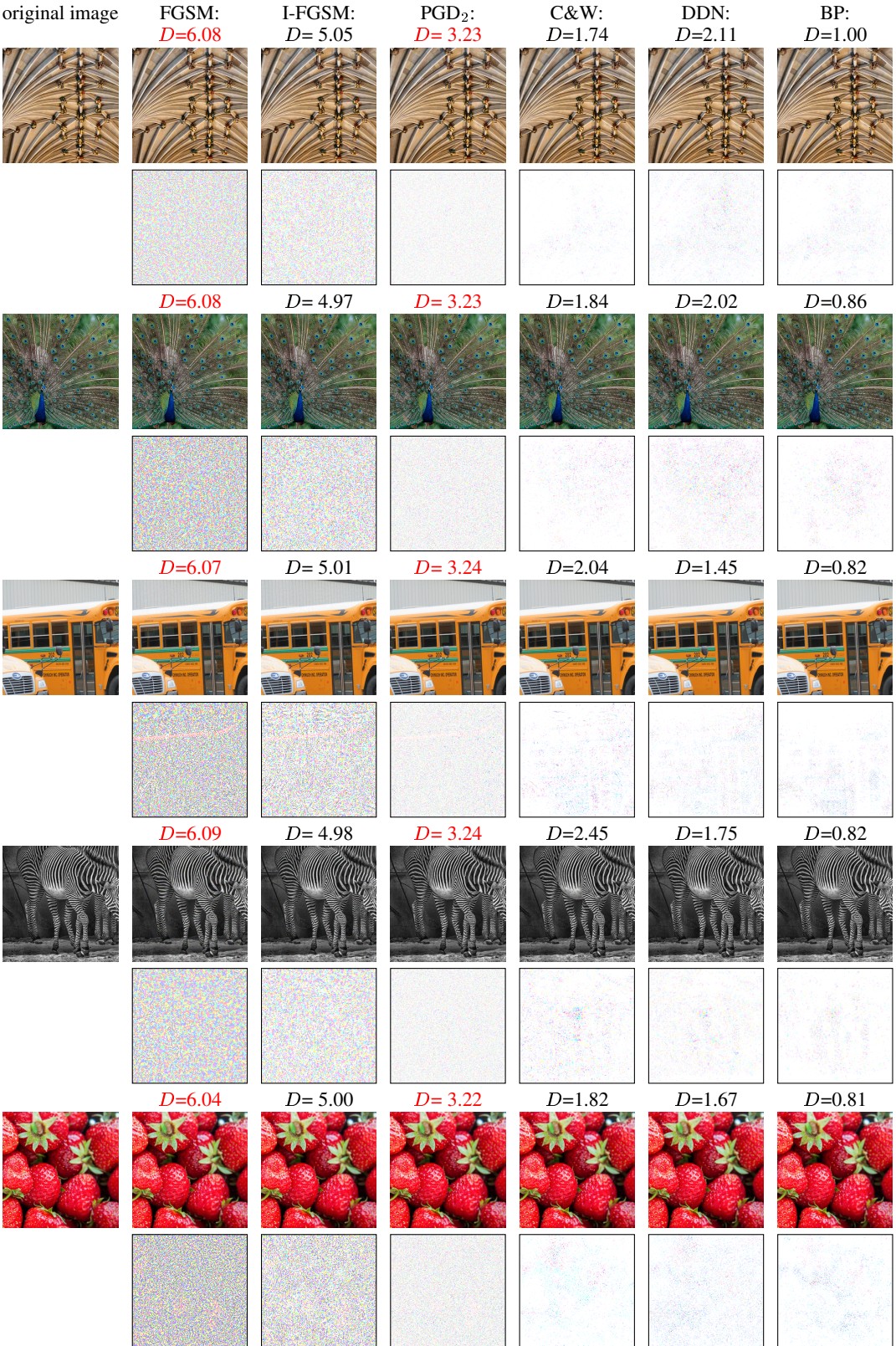

Figure 9: Original (left), adversarial (top row) and scaled perturbation (below) images against InceptionV3 on ImageNet. The five images are the worst 5 images for BP requiring the strongest distortions, yet these are smaller than the distortions necessary with all other methods (The red color means that the forged image is not adversarial). Perturbations are inverted (low is white; high is colored, per channel) and scaled in the same way for a fair comparison.

