# OpenReview forum: "Walking on the Edge: Fast, Low-Distortion Adversarial Examples"
_ICLR.cc/2020/Conference — Reject_

### Official Review · AnonReviewer1 · 2019-10-23
**Official Blind Review #1**

**Rating:** 3

**Review:**

This paper considers efficiently producing adversarial examples for deep neural networks and proposes boundary projection (BP), which quickly searches an adversarial example around the classification boundary. The BP approach is tested on three benchmark datasets and compared with existing adversarial attacking methods.

The key idea of BP, searching on the class boundary manifold, is interesting and promising. However, despite the excess of the recommended 8 pages, the main parts of the proposed method are not so clearly explained.

- It is not so clear which parts of the proposed method (Section 3) are mathematically justified. For example, \gamma_i in Eq. (14) looks heuristically introduced.
- Although the abstract and introduction emphasize that the main focus of BP is speed-distortion tradeoff, the experiments section does not discuss it so much and so clearly. While the operating characteristic of probability of success and distortion is mainly discussed, it is unclear which argument most demonstrate the improvement in speed-distortion tradeoff.

p.5, l.7: 1(a) -> Figure 1(a)
p.8, l.10: measure measure
p.8, right after Eq. (21): `"`is conditioned is" -> ``"is conditioned



**Experience Assessment:**

I do not know much about this area.

**Review Assessment: Checking Correctness Of Derivations And Theory:**

I assessed the sensibility of the derivations and theory.

**Review Assessment: Checking Correctness Of Experiments:**

I assessed the sensibility of the experiments.

**Review Assessment: Thoroughness In Paper Reading:**

I made a quick assessment of this paper.

---

> ### Author Response · Authors · 2019-11-11
> **Response to Reviewer 1**
>
> Thank you for your careful and valuable comments. We address your concerns point by point below.
>
> As general feedback, we fail to see how the concerns discussed here could support a "weak reject" recommendation, especially given the low confidence and quick assessment as stated. Could you please let us know if there are any other issues?
>
> Comment 1: "However, despite the excess of the recommended 8 pages, the main parts of the proposed method are not so clearly explained."
>
> Response 1: We have put a lot of effort in describing the method with motivation, description, pseudo-code and diagrams.
> The other two reviewers clearly understand our method. R3 even says the paper is a good read.
> Could you please elaborate on what is not clear? We are willing to make it as clear as possible.
>
>
> Comment 2: "It is not so clear which parts of the proposed method (Section 3) are mathematically justified. For example, \gamma_i in Eq. (14) looks heuristically introduced."
>
> Response 2: This concern is on using heuristics rather than on clarity.
>
> The $\gamma_i$ heuristic (14) builds on a simpler idea of DDN (Rony et al., 2019), where parameter $\gamma$ is constant across iterations.
>
> As explained, $\gamma_i$ controls the distortion:
> In stage 1, updates are small at the beginning to keep distortion low, then larger until the attack succeeds.
> In stage 2, updates are decreasing as $\gamma_i$ tends to 1. It increases the distortion when the current image is correctly classified (IN) and decreases the distortion when the current image is adversarial (OUT).
> All this behavior is controlled by a single parameter, which simplifies the algorithm.
>
> The fact that $(\gamma_i)_i$ is strictly increasing allows us to show that, in Stage 2, an IN iteration (distortion grows by $1/\gamma_i > 1$) followed by an OUT iteration (distortion decays by $\gamma_{i+1} < 1$) is indeed equivalent to a milder IN in the sense that the distortion grows by $\gamma_{i+1}/\gamma_i$ which is larger than 1 but smaller than $1/\gamma_i > 1$. Similarly, OUT followed by IN is equivalent to a mild OUT in the sense that distortion decays by $\gamma_i/\gamma_{i+1} < 1$. Both cases lead towards the class boundary by a factor that tends to 1: if the algorithm keeps alternating between OUT and IN and we only look at the OUT iterates (remember, all attacks output the successful iterate of least distortion), this is equivalent to strictly decreasing distortion. This behavior is more stable than having a constant parameter $\gamma$ as in DDN. We shall add this jusitification.
>
> From all the possible increasing sequences $(\gamma_i)_i$ that go to 1 as i goes to the maximum number of iterations, we pick the simplest one: a linear sequence. That is the only heuristic.
>
>
> Comment 3: “Although the abstract and introduction emphasize that the main focus of BP is speed-distortion tradeoff, the experiments section does not discuss it so much and so clearly. While the operating characteristic of probability of success and distortion is mainly discussed, it is unclear which argument most demonstrates the improvement in speed-distortion tradeoff.”
>
> Response 3: The speed-distortion trade-off is partially addressed by reporting results for 20 and 100 iterations in Tables 1, 2, 3. A more complete treatment is given in Appendix B.1 and Fig. 5, showing probability of success and distortion for more choices than 20 and 100 iterations.
> Following the benevolent recommendation of R3, we will move Appendix B.1 and Fig. 5 to the main body of the paper.
> As attacks become more and more powerful, speed becomes as important as distortion and probability of success.
>
>
> Comment 4:
> p.5, l.7: 1(a) -> Figure 1(a);
> p.8, l.10: measure measure;
> p.8, right after Eq. (21): `"`is conditioned is" -> ``"is conditioned"
>
> Response 4: Thank you for spotting these mistakes. They are now corrected.

---

### Official Review · AnonReviewer2 · 2019-10-25
**Official Blind Review #2**

**Rating:** 3

**Review:**

This paper proposed an adversarial attack method based on optimization on the manifold. The authors claim it is a fast and effective attack even with quantization.

It would better to also evaluate the method on the state of the art robust models (such as Madry et al ICLR'18) instead of only testing it on natural models. Generating adversarial examples on natural models is rather a well-solved problem and I do not think a 0.1 decrease in L2 norm is a big contribution since it is already so small that humans cannot distinguish. A better way to prove the strength would be to test it on a robust model to achieve higher success rates given a maximum distortion.

I do not think the results in Table 3 are convincing or necessary. It is well-known that the FGSM is so weak that the adversarial examples produced by it are not strong enough for adversarial training. The state of the art adversarial training defense uses the adversarial examples obtained from PGD. Also, a popular way to evaluate model robustness would be to evaluate the attack success rate under a given upper bound of distortion (e.g. 0.3 for MNIST). If there is no constraint on the distortion, we can always achieve a 100% attack success rate by simply use an image from another class. So in Table 3, the authors may either make sure all attacks have a 100% success rate and compare the distortion, or set an upper bound of distortion and compare the success rate (just as in the operating characteristics plot). With the current results, I do not believe the robust training with BP can be any better than FGSM. Similar issues also exist in Table 2.

**Experience Assessment:**

I have published one or two papers in this area.

**Review Assessment: Checking Correctness Of Derivations And Theory:**

I assessed the sensibility of the derivations and theory.

**Review Assessment: Checking Correctness Of Experiments:**

I assessed the sensibility of the experiments.

**Review Assessment: Thoroughness In Paper Reading:**

I read the paper at least twice and used my best judgement in assessing the paper.

---

> ### Author Response · Authors · 2019-11-11
> **Response to Reviewer 2 (part 1/2)**
>
> Thank you for your careful and valuable comments. We address your concerns point by point.
>
> Comment 1: “It would better to also evaluate the method on the state of the art robust models (such as Madry et al ICLR'18) instead of only testing it on natural models.”
>
> Response 1: All attacks keep reporting performance on natural models. For completeness, this is the first kind of evaluation every attack should have.
> Moreover, we do compare to robust models too. And not just that; we also use our attack to build an even more robust model.
>
> Table 3 compares three robust models obtained by adversarial training, each using a different attack including ours.
> As a defense, our attack (BP) has a similar or better performance than DDN. According to (Rony et al., 2019), the model obtained by adversarial training based on the DDN attack beats the robust model of (Madry et al, ICLR’18). Moreover, the training process of DDN is more efficient since standard (clean) model training is followed by few epochs using adversarial examples alone; while (Madry et al ICLR’18) train from scratch using a mix of clean and adversarial examples. This is why we have chosen DDN as a state of the art defense, which we further improve with our BP defense.
>
> Comment 2: “I do not think the results in Table 3 are convincing or necessary. It is well-known that the FGSM is so weak that the adversarial examples produced by it are not strong enough for adversarial training. The state of the art adversarial training defense uses the adversarial examples obtained from PGD. ... With the current results, I do not believe the robust training with BP can be any better than FGSM. Similar issues also exist in Table 2.”
>
> Response 2: According to Table 3, adversarial training with our BP is similar or better than DDN in terms of distortion, and it also beats adversarial training with FGSM by a large margin in all cases (higher distortion for all attacks). In fact, FGSM defense was included as a baseline since it was the first method used for adversarial training. We agree with the reviewer's comment that FGSM is weak.
>
> Besides, since our attack is fast, the adversarial training is fast. Table 3 is then necessary, showing improvements in defense over DDN, which in turn improves over PGD (Madry et al, ICLR'18), which in turn improves over FGSM. For completeness, we shall add PGD and the original (Madry et al ICLR’18) defenses to make Table 3 more convincing, as well as corresponding operating characteristics like those of Fig. 3.

---

> ### Author Response · Authors · 2019-11-11
> **Response to Reviewer 2 (part 2/2)**
>
> Comment 3: “Also, a popular way to evaluate model robustness would be to evaluate the attack success rate under a given upper bound of distortion (e.g. 0.3 for MNIST). If there is no constraint on the distortion, we can always achieve a 100% attack success rate by simply use an image from another class..”
>
> Response 3: This comment, in fact, applies equally to all our results in Tables 1-3, so we discuss it in this general context.
>
> This kind of evaluation is popular indeed, but is biased to target distortion attacks.
> As explained in Section 2.2, target distortion attacks address the first form (2-3) of the problem: maximize the success rate subject to the upper bound of distortion. Our algorithm is clearly a target success attack addressing the second form (4-5) of the problem: minimize distortion subject to attack success. Hence our evaluation in Tables 1-3 is biased to this second form, having attack rates close to 1 and measuring quality in terms of distortion.
> (Note that not all target success attacks have a 100\% success rate because of the additional constraint in complexity as given by the limited number of iterations.)
>
> A fair comparison of both types of attacks is tricky and cannot be done as proposed by the reviewer. For this reason, Section 4.2 explains a new protocol, also introducing operating characteristics as in Fig. 3. This protocol is actually another contribution to our work. It unifies both forms into a single benchmark, facilitating fair comparisons of target distortion attacks (FGSM, I-FGSM, PGD) and target success attacks (C&W, DDN, BP). To evaluate target distortion attacks, we vary the upper bound epsilon and take successful adversarial examples of minimal distortion. This applies to all measurements (statistics and operating characteristics).
>
> By taking a fixed distortion (x-axis) in Figure 3, the characteristic function gives a probability of success (y-axis) equal *by definition* to the attack success rate under a given upper bound on distortion, as suggested by the reviewer. Similarly, by taking a fixed probability of success (y-axis), the characteristic function gives a distortion (x-axis) equal to the least upper bound of distortion required to achieve such a success rate.
>
> So, what the reviewer is recommending was already contained in the operating characteristic. To facilitate its reading, we will extract success rates for a given upper bound of distortion from the operating characteristics and report them in Tables 1-3 as recommended. As we strongly believe that operating characteristic is the only fair comparison, we will add the corresponding ones of Table 3 as stated in points 1,2 above.
>
>
> Comment 4: “Generating adversarial examples on natural models is rather a well-solved problem and I do not think a 0.1 decrease in L2 norm is a big contribution since it is already so small that humans cannot distinguish.”
>
> Response 4: This statement is questionable from many perspectives.
>
> Number 0.1 is used here as an example of a small number without any reference measurement. For instance, our improvement over DDN on ImageNet (the most realistic dataset) at 100 iterations is 0.15 (0.43 -> 0.28), a relative decrease of -34%. More importantly, at 20 iterations, the improvement is 0.83 (1.18 -> 0.35), an impressive relative decrease of -70%.
>
> We agree that humans may not distinguish so low distortions. Yet, since operating characteristics rarely cross, an attack with lower average distortion has in general higher success rate for a given upper bound of distortion. In other words, improvements in distortion (subject to success) are in general equivalent to improvements in success rate (under an upper bound of distortion).
>
> Is generating adversarial examples a well-solved problem? Not really if one constrains the number of iterations, which is central to our approach. Very recent attacks keep reporting improvements in distortion (in natural models included) with hundreds of thousands of iterations. This is not great progress. As far as we know, DDN is the only state of art attack that produces adversarial examples with low distortion at few iterations, allowing its use in adversarial training. Our attack further improves this state of the art.
>
> Is generating adversarial examples a well-solved problem? Not really if one takes into account quantization. Images are quantized in the real world, but almost all academic papers do not consider this constraint. Quantization jeopardizes the adversarial perturbation especially at small distortion: After quantization, some real matrices are no longer adversarial. However, only we and DDN (Rony et al., 2019) consider this effect. This is the reason why we evaluate different attacks with quantization (Table 2) and without quantization (Table 4). It turns out that our attack works well within both cases.

---

### Official Review · AnonReviewer3 · 2019-10-29
**Official Blind Review #3**

**Rating:** 6

**Review:**

This paper introduces a parameterized approach to generate adversarial samples by balancing the speed-distortion trade-off. The method first tries to reach the boundary of classes in the classifier space, then walks on the classifier manifold to find adversarial samples that make the classifier to fail in prediction while minimizing the level of distortion in the sample. Having a limited number of iterations, the method reduces the fluctuations around the boundary and paves the classification manifold.

The idea is novel, interesting and well-formulated, while the intuition could be better explained. The paper is a good read, has an adequate amount of literature review, and the results are supporting the claims of the paper: lower distortion while having comparable accuracy, the use of generated samples in fortifying the classifier, and keeping distortion to a reasonable level (qualitative results in appendix). However, one of the claims is to trade the distortion level to speed that needs verifying in the main manuscript, therefore, it is suggested that the section B.1 moves to the main manuscript and discussed more thoroughly. Also the effect of other parameters on this trade-off (such as the number of iterations K).

It is also interesting to discuss how the algorithm performs in classes that are linearly separable on a toy dataset.

**Experience Assessment:**

I have read many papers in this area.

**Review Assessment: Checking Correctness Of Derivations And Theory:**

I assessed the sensibility of the derivations and theory.

**Review Assessment: Checking Correctness Of Experiments:**

I carefully checked the experiments.

**Review Assessment: Thoroughness In Paper Reading:**

I read the paper thoroughly.

---

> ### Author Response · Authors · 2019-11-11
> **Response to Reviewer 3**
>
> Thank you for your careful and valuable comments. We address your concerns point by point.
>
> Comment 1: “However, one of the claims is to trade the distortion level to speed that needs verifying in the main manuscript, therefore, it is suggested that the section B.1 moves to the main manuscript and discussed more thoroughly.”
>
> Response 1: It is reasonable to move section B.1 to the main manuscript. This should also help with respect to point 3 of reviewer 1. We will find the space for it and update the manuscript, but this probably means moving some other material to an Appendix or otherwise significantly shortening or removing other material.
>
>
> Comment 2: “Also the effect of other parameters on this trade-off (such as the number of iterations K).”
>
> Response 2: The effect of the number of iterations $K$ is exactly the same as $\# grad$ as shown in Fig. 5 because we only calculate one gradient per iteration.
> Other parameters are $\alpha$ and $\gamma_{min}$. We will add more results to show the effect of these parameters in an appendix.
>
>
> Comment 3: “It is also interesting to discuss how the algorithm performs in classes that are linearly separable on a toy dataset.”
>
> Response 3: Considering Fig. 1, if the classes were linearly separable, the boundary and all iso-contours would be straight lines, the gradient would be normal to these lines, and all algorithms would move along a line normal to the boundary. The problem would then be one-dimensional, which is not interesting to display like Fig. 1. What may be more interesting is to plot the position on this line as a function of iteration. This we may attempt to include in an appendix.

---

### Author Response · Authors · 2019-11-15
**Summary to all reviewers and paper updates**

This is a response to all reviewers, meant to summarize the main points and the updates we have made to the paper.

1. We would like to thank all reviewers for their in-depth feedback. As a result of this discussion, we are improving a lot our paper. For each point below, we discuss the corresponding updates we have done or we shall do in the paper. Most new material has now been added in the appendices. In the final paper we will re-organize.


2. We are already evaluating attacks against robust models including adversarial training by DDN, which is superior than (Madry et al, ICLR’18), in Table 3.

Update of the paper:

We now include in Table 5 attack evaluation against robust models including (Madry et al, ICLR’18) on MNIST and CIFAR10 as well as (Tramer et al., 2017) on ImageNet. This replaces a Table in the previous version that was evaluating attacks against classifiers adversarially trained against FGSM. Our BP is still the strongest attack in general.


3. We are evaluating distortion for given success rate close to 1 because BP is a target distortion attack. However, we also introduce a new protocol that facilitates fair comparison of target distortion with target success attacks, including operating characteristics. This allows any kind of measurement and shows that in general, an attack with smaller average distortion will have higher success rate for given upper bound on distortion.

Update of the paper:

For completeness, we now define $P_{\texttt{upp}}$ which is the success rate when the distortion is constrained $D(x) < D_{\texttt{upp}}$. It amounts to evaluating the operating characteristic $D\rightarrow P(D)$ at some $D = D_{\texttt{upp}}$. This statistic, called "success rate", is added in all tables next to probability of success and average distortion. We choose a small distortion, highlighting the fact that at low distortion our method outperforms the others by a large margin.

We now also include operating characteristc plots (Fig. 7 and 8) for attack evaluation without quantization and against robust models. This offers a unified view of all metrics and shows that our BP is the strongest attack in all experiments. It is close to DDN in many cases, because the plots are for 100 iterations. At 20 iterations, BP is even better.


4. We study and improve upon the quality of adversarial examples under limited iterations and quantization as part of the evaluation. This is ignored in most of the literature. In this sense, even attacking natural models is not well-solved.

Update of the paper:

The difference is evident now by comparing Fig. 3 (with quantization) to the new Fig. 7 (without quantization).


5. We provide more information on selected parameters.

Update of the paper:

In Appendix B.1, we now study the impact of the attack parameter: $\alpha$ and $\gamma_{\min}$.
It turns out that performance is stable wrt $\alpha$ provided it is large enough. As for $\gamma_{min}$, the best value is 0.7 whatever the value of $\alpha$.


6. We shall include a justification of the schedule of $\gamma_i$ (14) summarizing the discussion we had with Reviewer 1 on the behavior of the algorithm.


7. We shall move the "speed - distortion trade-off" (now appendix B.2) to the main paper.

---

### Decision · Program_Chairs · 2019-12-19

**Decision:**

Reject

**Comment:**

In this paper the authors highlight the role of time in adversarial training and study various speed-distortion trade-offs. They introduce an attack called boundary projection BP which relies on utilizing the classification boundary. The reviewers agree that searching on the class boundary manifold, is interesting and promising but raise important concerns about evaluations on state of the art data sets. Some of the reviewers also express concern about the quality of presentation and lack of detail. While the authors have addressed some of these issues in the response, the reviewers continue to have some concerns. Overall I agree with the assessment of the reviewers and do not recommend acceptance at this time.